# Knowledge Swapping via Learning and Unlearning

**Mingyu Xing** [1]  **Lechao Cheng**[✉ 1]  **Shengeng Tang** [1]  **Yaxiong Wang** [1]  **Zhun Zhong**[✉ 1 2]  **Meng Wang** [1]

## Abstract

We introduce **Knowledge Swapping**, a novel task designed to selectively regulate knowledge of a pretrained model by enabling the forgetting of user-specified information, retaining essential knowledge, and acquiring new knowledge simultaneously. By delving into the analysis of knock-on feature hierarchy, we find that incremental learning typically progresses from low-level representations to higher-level semantics, whereas forgetting tends to occur in the opposite direction—starting from high-level semantics and moving down to low-level features. Building upon this, we propose to benchmark the knowledge swapping task with the strategy of *Learning Before Forgetting*. Comprehensive experiments on various tasks like image classification, object detection, and semantic segmentation validate the effectiveness of the proposed strategy. The source code is available at https://github.com/xingmingyu123456/Knowledge Swapping

## 1. Introduction

The proliferation of deep learning has led to the widespread adoption of pretrained models (Han et al., 2021), which are typically fine-tuned using task-specific data to achieve parameter-efficient adaptation. In the context of streaming tasks, researchers have increasingly explored methods for continuously optimizing and adapting pretrained models to new tasks (Lin et al., 2023; Chen et al., 2024; Zhu et al., 2023), a paradigm referred to as continual learning (Wang et al., 2024). However, in practical applications, as deep models integrate more knowledge, they often encounter additional requirements, such as the need to continuously

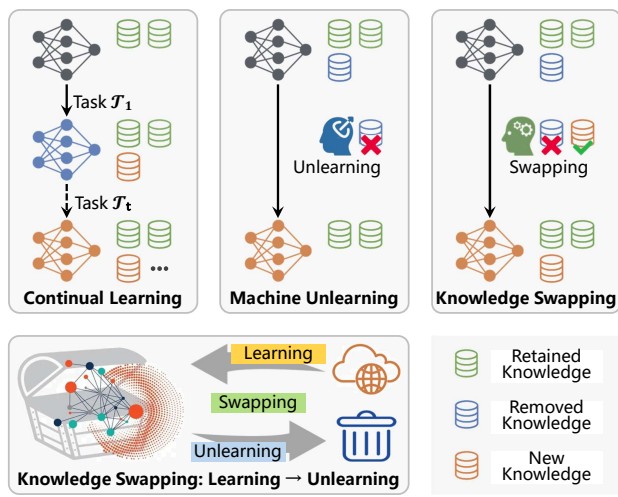

*Figure 1.* Comparisons of three tasks: Continuous Learning, Machine Unlearning, and our Knowledge Swapping.

block or forget certain sensitive content. Recent works, such as machine unlearning (Liu et al., 2024c; Wang et al., 2025; Zhang et al., 2024b), have begun to address the unilateral forgetting of specific content within pretrained models. Nonetheless, approaches that simultaneously enable the learning of new knowledge and the forgetting of specific content remain underexplored.

Inspired by this insight, we propose a novel task termed Knowledge Swapping, which enables the selective forgetting of specific categories of knowledge while preserving others, tailored to the requirements of new tasks. As depicted in Figure 1, *continual learning* (Cheng et al., 2024a;b; Zhang et al., 2024a; Huang et al., 2024a;b; Wu et al., 2025) aims to integrate new task knowledge into an existing pretrained model. Current mainstream approaches (Li et al., 2024; Wang et al., 2022b) often involve attaching a new adapter for each task and fine-tuning it, yet the need to forget outdated or less relevant knowledge remains an underexplored challenge. Recent advancements (Liu et al., 2024b; Wang et al., 2025) in *machine unlearning* have demonstrated that isolating or removing specific knowledge from a pretrained model necessitates an explicit unlearning process. In contrast, our proposed **Knowledge Swapping** task facili-

[1]School of Computer Science and Information Engineering, Hefei University of Technology [2]School of Computer Science, University of Nottingham. Correspondence to: Lechao Cheng <chenglc@hfut.edu.cn>, Zhun Zhong <zhunzhong007@gmail.com>.

*Proceedings of the 42nd International Conference on Machine Learning*, Vancouver, Canada. PMLR 267, 2025. Copyright 2025 by the author(s).

tates the learning of new tasks while simultaneously forgetting less important prior knowledge or sensitive data that requires protection, thereby preserving the core capabilities of the pretrained model.

We further delve into how to perform knowledge swapping more effectively by leveraging empirical insights. Intuitively, this process can be divided into two stages: forgetting specific knowledge and learning novel knowledge. A natural assumption might be that the model should first forget less important or potentially detrimental priors in order to "free up" the capacity for new information. We conduct a set of straightforward experiments to analyze, in isolation, how incremental learning and targeted forgetting each influence model parameters. Our findings indicate that incremental learning typically progresses from low-level representations to higher-level semantic features, whereas forgetting tends to occur in the opposite direction—starting from high-level semantics and moving down to low-level features. This contrast provides valuable insights for designing strategies for knowledge swapping task. Specifically, if we perform targeted forgetting first, we may complete the forgetting of high-level semantics before any major adjustments occur in the low-level feature space. Once we begin introducing new content afterward, these low-level changes may no longer align with the previously forgotten high-level semantic representations, thereby creating potential conflicts. Conversely, if we first learn the new task (thus updating the low-level feature distribution), and only then conduct targeted forgetting, the process is more likely to remain confined to higher-level semantics. As a result, the updated low-level distributions are less likely to be perturbed during the forgetting stage, which helps to maintain the integrity of previously acquired knowledge. We empirically show *learning new tasks first, followed by selective forgetting of specific knowledge, leads to significantly better results.* Our contributions are summarized as follows:

- We propose the concept of **Knowledge Swapping**, a novel task that facilitates the learning of new tasks while simultaneously forgetting less important prior knowledge and preserving essential pretrained capabilities.

- We uncover that incremental learning progresses from low-level to higher-level semantic features, whereas targeted forgetting begins at high-level semantics and works downward. This directional contrast provides key insights into how to design effective knowledge swapping procedures.

- Building upon the insight from feature-hierarchy interplay, we propose to achieve knowledge swapping by sequential learning-then-forgetting principle. Comprehensive experiments also demonstrate that the pro-

posed strategy significantly improves overall performance.

## 2. Related Work

### 2.1. Continual Learning

Continual learning, also known as lifelong learning, aims to enable models to learn new tasks incrementally while mitigating catastrophic forgetting. Existing approaches can be broadly categorized into Regularization-Based Methods (Kirkpatrick et al., 2017; Li & Hoiem, 2017; Zhu et al., 2021; Liu & Liu, 2022; Wang et al., 2023), Memory-Based Methods (Rebuffi et al., 2017; Ashok et al., 2022; Chaudhry et al., 2021; Zhou et al., 2022; Sun et al., 2023; Yang et al., 2023), and Architecture-Based Methods (Rusu et al., 2016; Douillard et al., 2022; Wang et al., 2022a; Lu et al., 2024).

Regularization-based methods introduce constraints in the loss function to retain past knowledge. For instance, EWC (Kirkpatrick et al., 2017) distills knowledge from old models to new ones for prediction consistency. Memory-based methods maintain an external memory to store or generate past knowledge. BMKP (Sun et al., 2023) introduces a bilevel memory framework, where a working memory adapts to new tasks and a long-term memory retains compact knowledge representations. Architecture-based methods expand the model to accommodate new tasks. Progressive Neural Networks(Rusu et al., 2016) expands the model using gradient boosting and compresses it via knowledge distillation. ArchCraft (Lu et al., 2024) leverages neural architecture search (NAS) to balance stability and plasticity, generating architectures that enhance knowledge retention with minimal parameter overhead. Our proposed method shares the goal of balancing retention and learning with continual learning approaches. However, knowledge swapping introduces an additional challenge of selective forgetting, which is not explicitly addressed in traditional continual learning methods.

### 2.2. Machine Unlearning

Machine unlearning (Koh & Liang, 2017; Kurmanji et al., 2024; Liu et al., 2024a) focuses on efficiently removing specific data or knowledge from a trained model without requiring full retraining, which is crucial for data privacy compliance. One of the earliest approaches is fine-tuning, which exploits catastrophic forgetting by retraining the model on a retention dataset, though it may leave residual traces of forgotten data. This method forms the foundation for subsequent unlearning techniques. Building on this, Influence Functions (Koh & Liang, 2017) emerged, whiestimatetes the influence of individual data points on model parameters, providing a more precise and computationally efficient method for data removal without retraining the entire model. Later,

more sophisticated methods are introduced. NegGrad+ (Kurmanji et al., 2024) balances the loss on the forgotten set and the retention dataset, offering a more controlled trade-off in the unlearning process. To further refine the removal of specific knowledge, L1-Sparse (Liu et al., 2024a) introduces the use of L1-regularization to zero out parameters associated with forgotten data, effectively eliminating their influence on the model. Additionally, Relabeling techniques, such as Saliency Unlearning (Feldman, 2020) disrupts the model's memory of forgotten data by altering its labels, focusing on modifying key parameters that store the data's influence. Unlike conventional unlearning methods, which primarily focus on individual data points, our proposed framework introduces a novel approach for category-level forgetting. By integrating the processes of learning, retention, and forgetting into a unified system, this framework offers more flexibility and control over knowledge management, marking a significant advancement in this area of research.

## 3. Knowledge Swapping

### 3.1. Task Definition

We introduce a novel task, referred to as **Knowledge Swapping**, which aims to selectively regulate a model's knowledge by employing specific swapping mechanisms to achieve three primary objectives: (1) forgetting user-specified knowledge, (2) retaining core knowledge, and (3) simultaneously learning new knowledge.

Let $\mathcal{D}_p = (X_p, Y_p)$ be the **pretraining dataset** on which an initial model $M_0$ has been trained. We define three additional sets:

- **Retaining Set:** $\mathcal{D}_r = (X_r, Y_r)$, which contains the knowledge that must be preserved.

- **Forgetting Set:** $\mathcal{D}_f = (X_f, Y_f)$, containing the knowledge that the model needs to forget.

- **Learning Set:** $\mathcal{D}_l = (X_l, Y_l)$, comprising new knowledge to acquire.

In general, both $\mathcal{D}_r$ and $\mathcal{D}_f$ are subsets of the pretraining dataset $\mathcal{D}_p$, i.e., $\mathcal{D}_r, \mathcal{D}_f \subseteq \mathcal{D}_p$, while $\mathcal{D}_l$ could represent an entirely new task or domain.

**Objectives:** We seek to train an updated model $M_1$ such that:

1. For each $x_r \in X_r$, $M_1$ *correctly* predicts its label $y_r$. Formally, $P(M_1(x_r) = y_r) \approx 1, \quad \forall x_r \in X_r$. (Retention)

2. For each $x_f \in X_f$, $M_1$ does *not* correctly predict its label $y_f$. Formally, $P(M_1(x_f) = y_f) \approx 0, \quad \forall x_f \in X_f$. (Forgetting)

3. For each $x_l \in X_l$, $M_1$ *correctly* predicts its label $y_l$. Formally, $P(M_1(x_l) = y_l) \approx 1, \quad \forall x_l \in X_l$ (Learning)

Here, $P(\cdot)$ denotes the probability of the corresponding event under the trained model. The objectives are then to promote or discourage each of these criteria accordingly.

### 3.2. To Forget Before Learning, or to Learn Before Forgetting?

Revisiting that we have already provided a comprehensive definition of the knowledge swapping task, we note that—based on its criteria—it can be naturally divided into two core phases: forgetting and learning. This highlights a pivotal question: should a model first forget certain existing knowledge before learning new information (which seems intuitively appealing), or should it learn new content first and then perform forgetting? The way we answer this sequence dilemma directly informs how we design a robust knowledge swapping benchmark. Although intuitively, forgetting old knowledge first appears to free up capacity for accommodating the new, is this really the optimal approach? To explore this, we conduct two sets of experiments on dense prediction tasks such as semantic segmentation, each corresponding to one of these two learning orders. We then examine how the neural network's parameters, across various layers, evolve under each approach (see Figure 2).

**Knock-on Feature Hierarchy** In this section, we directly evaluate the weight norms and parameter differences at each stage of the process. Specifically, $L \rightarrow F$ denotes *Learning Before Forgetting*, while $F \rightarrow L$ indicates *Learning After Forgetting*. The superscript $W$ represents the weight norms at different stages under each sequence. We aggregate results from multiple image segmentation tasks and observe that when following the $L \rightarrow F$ sequence, the majority of parameter updates occur in the latter layers of the neural network—those responsible for generating semantic-level features. Conversely, in the $F \rightarrow L$ sequence, the principal changes are concentrated in the earlier layers, which produce low-level features. Based on these results, we found:

**Discovery-I:** *Incremental learning typically progresses from low-level representations to higher-level semantic features, whereas forgetting tends to occur in the opposite direction—starting from high-level semantics and moving down to low-level features.*

**Remark.** What is the practical significance of this seemingly intuitive finding? Clearly, its hierarchical feature implications offer valuable insights for designing knowledge swapping strategies. Specifically, if we conduct targeted forgetting first, we may erase high-level semantic parameters before making any substantial adjustments to the low-level feature parameter space. However, once we subsequently

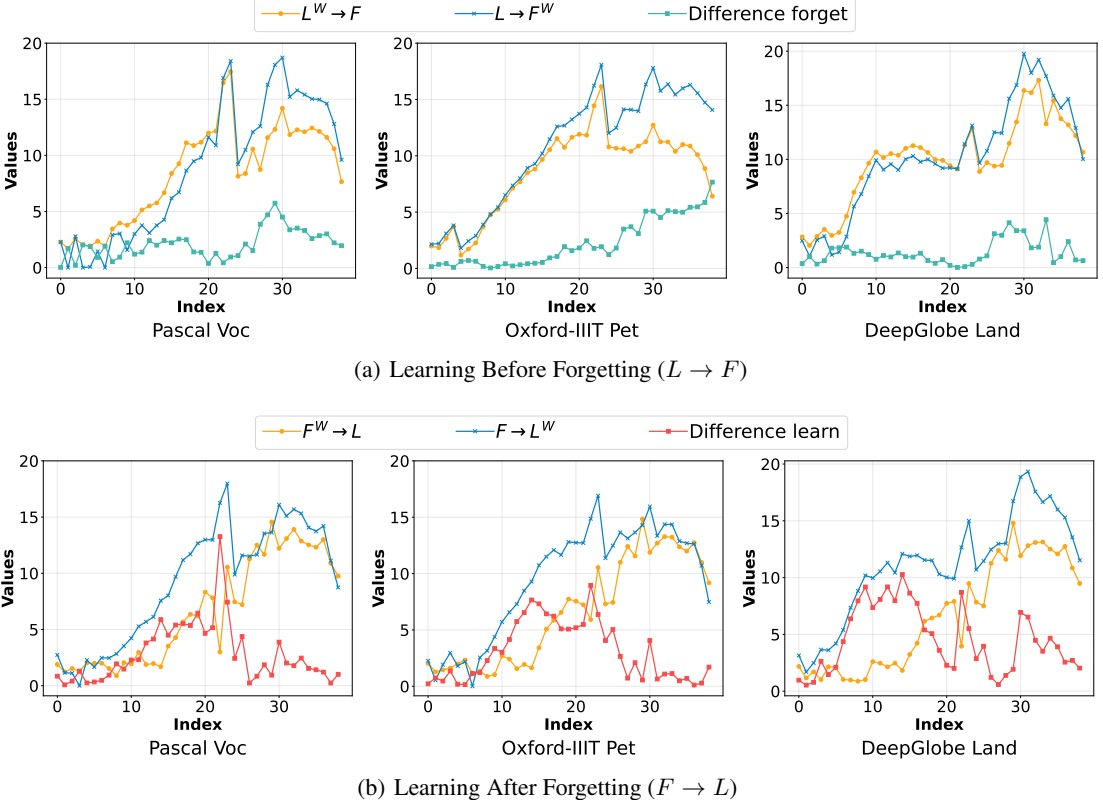

(a) Learning Before Forgetting ($L \rightarrow F$)

(b) Learning After Forgetting ($F \rightarrow L$)

*Figure 2.* $\mathcal{L}_2$ **norm for each parameter under** $L \rightarrow F$ **and** $F \rightarrow L$. The superscript $W$ denotes the weight norm value at the current stage. The figure illustrates that (a) during the *Learning Before Forgetting* phase, changes in parameter norms are predominantly concentrated in layers responsible for high-level semantic representations. Conversely, (b) in the *Learning After Forgetting* phase, parameter norm changes primarily occur in layers associated with low-level feature representations.

introduce new content, further modifications to the low-level parameters can cause inconsistencies in the high-level semantics (breaking the forgetting pipeline owing to altering low-level parameters), potentially leading to conflicts. By contrast, if we begin by learning a new task (thereby updating the low-level parameters) and then perform targeted forgetting, the forgetting process is more likely confined to the high-level semantic parameters.

**Principle: Learning Before Forgetting.** Recall that we demonstrate that initiating the forgetting process before learning disrupts the intended forgetting due to alterations in low-level feature learning. This raises the question: does conducting the learning process before forgetting similarly interfere with previously acquired knowledge? We also measure the average gradient changes across two primary sequences, $L \rightarrow F$ and $F \rightarrow L$, as shown in Figure 4. The superscript $G$ means the log average gradient at the current stage. First, parameter changes during the learning phases ($L^G \rightarrow F$ and $F \rightarrow L^G$) are consistently more significant, indicating that the *Learning* process is relatively challenging; second, in the $L \rightarrow F^G$ phase, the final updates to forgetting gradients remain consistently small, suggesting

that *Learning Before Forgetting* is more stable.

## 4. Benchmark

### 4.1. Overview

As illustrated in Figure 3, we employ the Low-Rank Adaptation (LoRA) technique (Hu et al., 2021) to fine-tune the pretrained model $M_0$ ( Sec. 4.2). Additionally, we leverage group sparse regularization to constrain the selective learning and forgetting of specific knowledge (Sec. 4.3).

### 4.2. LoRA-Based Fine-Tuning

Building on the findings in (Geva et al., 2020), which demonstrate that the linear layers within Transformers encapsulate a substantial amount of the model's knowledge, we employ the Low-Rank Adaptation (LoRA) technique (Hu et al., 2021) to fine-tune only these linear layers. Let $\boldsymbol{X}$ denote the input to the Feed-Forward Network (FFN) of the $k$-th Transformer block after the $t$-th gradient update. The weights and biases of the first linear layer are represented by $\boldsymbol{W}_{k1}^t$ and $\boldsymbol{b}_{k1}^t$, respectively, while those of

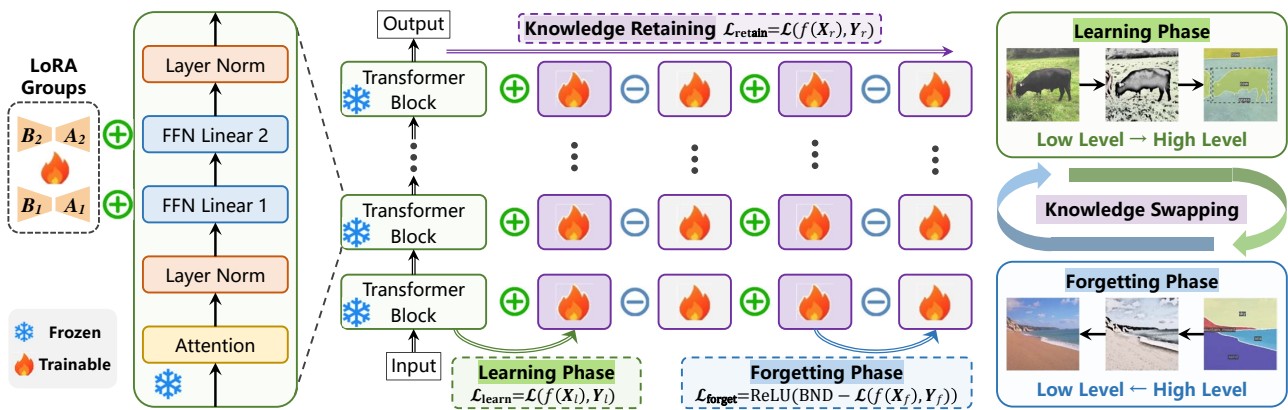

*Figure 3.* Benchmark Framework. First, we decouple knowledge swapping into separate learning and forgetting processes. We observed that the learning process progresses from low-level features to high-level features, while the forgetting process proceeds in the opposite direction—from high-level features to low-level features. Therefore, a two-stage strategy of *Learning Before Forgetting* is adopted. In general, we adopt LoRA to fine-tune the linear layers in each Transformer block, with all other parameters frozen to enable selective regulation of the model knowledge.

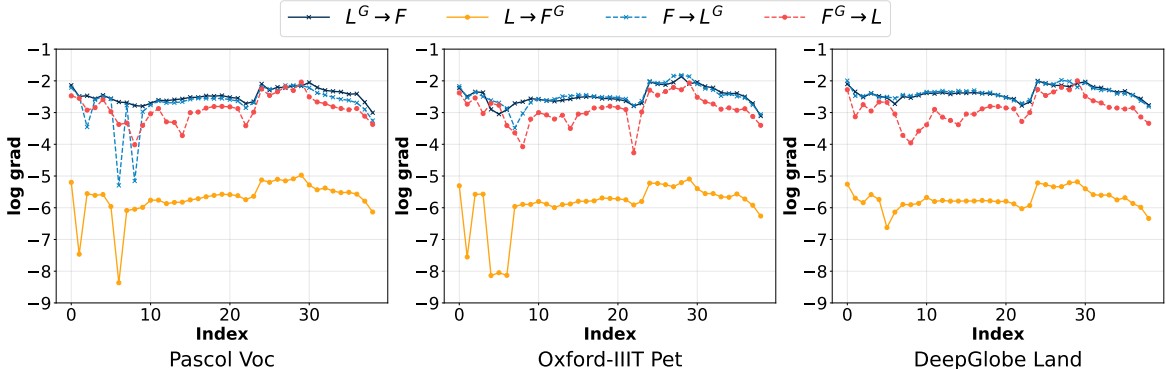

*Figure 4.* **Logarithm of the Average Gradient.** We compute the logarithm of cumulative average gradient changes at different stages in the $L \to F$ and $F \to L$ processes. We observe two key phenomena: first, parameter changes during the learning phases ($L^G \to F$ and $F \to L^G$) are consistently more significant, indicating that the *Learning* process is relatively challenging; second, in the $L \to F^G$ phase, the final updates to forgetting gradients remain consistently small, suggesting that *Learning Before Forgetting* is more stable.

the second linear layer are denoted by $\boldsymbol{W}_{k2}^t$ and $\boldsymbol{b}_{k2}^t$. The computation performed by the FFN can be expressed as:

$$\text{FFN}_k^t(\boldsymbol{X}) = \text{ReLU}\left(\boldsymbol{X}\boldsymbol{W}_{k1}^t + \boldsymbol{b}_{k1}^t\right)\boldsymbol{W}_{k2}^t + \boldsymbol{b}_{k2}^t. \quad (1)$$

Using LoRA, the weights are decomposed into their original pretrained components and learnable low-rank adaptations:

$$\boldsymbol{W}_{k1}^t = \boldsymbol{W}_{k1}^0 + \Delta\boldsymbol{W}_{k1}^t, \quad \Delta\boldsymbol{W}_{k1}^t = \boldsymbol{A}_{k1}^t\boldsymbol{B}_{k1}^t. \quad (2)$$

$$\boldsymbol{W}_{k2}^t = \boldsymbol{W}_{k2}^0 + \Delta\boldsymbol{W}_{k2}^t, \quad \Delta\boldsymbol{W}_{k2}^t = \boldsymbol{A}_{k2}^t\boldsymbol{B}_{k2}^t. \quad (3)$$

In these equations, $\boldsymbol{W}_{k1}^0$ and $\boldsymbol{W}_{k2}^0$ represent the original pretrained weights, while $\Delta\boldsymbol{W}_{k1}^t$ and $\Delta\boldsymbol{W}_{k2}^t$ are the low-rank updates at time step $t$, parameterized by the matrices $\boldsymbol{A}$ and $\boldsymbol{B}$. This approach ensures efficient adaptation by focusing solely on the linear layers of the Transformer, which are hypothesized to store the majority of the model's knowledge.

Although the biases $\boldsymbol{b}_{k1}^t$ and $\boldsymbol{b}_{k2}^t$ may also be fine-tuned, they typically involve fewer parameters compared to the weights. By restricting updates to low-rank matrices, LoRA enables efficient fine-tuning with reduced computational and storage overhead while preserving the knowledge embedded in the original pretrained weights.

### 4.3. Sparse Constriant

We employ Group Coefficient Regularization to selectively retain specific knowledge within the Feed-Forward Network (FFN) modules. Specifically, we adopt the Lasso strategy (Liu et al., 2015; Wen et al., 2016; Yuan & Lin, 2006) for group sparse regularization. Lasso achieves the selective retention of knowledge by zeroing out the $A$ and $B$ matrices of irrelevant FFN modules within LoRA, thereby enabling

*Table 1.* **Image classification results on Imagenet100 under *Learning Before Forgetting*.** $L$ and $F$ are short for *Learning* and *Forgetting*, respectively. $\text{Acc}_r$, $\text{Acc}_l$ and $\text{Acc}_f$ represent the accuracy of the retaining set, the learning set, and the forgetting set.

| Stages | Cub | | | Resisc45 | | | Oxford-pet | | | Plantvillage | | |
|---|---|---|---|---|---|---|---|---|---|---|---|---|
| | $\text{Acc}_r\uparrow$ | $\text{Acc}_l\uparrow$ | $\text{Acc}_f\downarrow$ | $\text{Acc}_r\uparrow$ | $\text{Acc}_l\uparrow$ | $\text{Acc}_f\downarrow$ | $\text{Acc}_r\uparrow$ | $\text{Acc}_l\uparrow$ | $\text{Acc}_f\downarrow$ | $\text{Acc}_r\uparrow$ | $\text{Acc}_l\uparrow$ | $\text{Acc}_f\downarrow$ |
| | *Learning Set*: 5 classes, *Forgetting Set*: 5 classes, *Retaining Set*: 95 classes | | | | | | | | | | | |
| Start | 88.08 | 0 | 84.4 | 88.08 | 0.2 | 84.4 | 88.08 | 0.4 | 84.4 | 88.08 | 0 | 84.4 |
| $F$ | 86.63 | 0 | 0.4 | 86.63 | 0.4 | 0.4 | 86.63 | 0.4 | 0.4 | 86.63 | 0 | 0.4 |
| $F \rightarrow L$ | 86.94 | 94.04 | 80.8 | 87.62 | 99 | 70.8 | 88.48 | 93.6 | 77.2 | 88.06 | 99.13 | 74.8 |
| $L$ | 86.67 | 94.04 | 82.44 | 87.93 | 98.8 | 82.0 | 87.83 | 94.4 | 82.8 | 87.97 | 99.56 | 82.4 |
| $L \rightarrow F$ | 86.4 | 91.66 | 0 | 86.77 | 99.2 | 0 | 87.01 | 93.6 | 0 | 87.26 | 99.78 | 0 |
| | *Learning Set*: 10 classes, *Forgetting Set*: 10 classes, *Retaining Set*: 90 classes | | | | | | | | | | | |
| Start | 87.75 | 0 | 89.2 | 87.75 | 0.1 | 89.2 | 87.75 | 1.8 | 89.2 | 87.75 | 0 | 89.2 |
| $F$ | 85.22 | 0 | 0 | 85.22 | 0.2 | 0 | 85.2 | 2 | 0 | 85.22 | 0 | 0 |
| $F \rightarrow L$ | 86.84 | 96.27 | 82 | 87.2 | 98.7 | 82 | 88 | 96 | 83 | 87.66 | 98.55 | 82.2 |
| $L$ | 85.06 | 95.03 | 83.6 | 86.84 | 98.9 | 84.4 | 87.0 | 95.8 | 85.2 | 87.35 | 98.86 | 85.4 |
| $L \rightarrow F$ | 84.68 | 93.78 | 0 | 85.97 | 98.7 | 0.6 | 86.22 | 95.2 | 0.2 | 85.48 | 98.65 | 0 |

targeted learning and forgetting. The Lasso regularization loss $\mathcal{L}_{\text{re}}$ is defined as:

$$\mathcal{L}_{\text{re}} = \sum_{k=1}^{n} \left( \|\boldsymbol{A}_k\|_F^2 + \|\boldsymbol{B}_k\|_F^2 \right), \tag{4}$$

where $\| \cdot \|_F$ denotes the Frobenius norm, calculated as the square root of the sum of the squared elements of a matrix, and $n$ represents the number of FFN groups.

**Remark.** Sparse constraints enhance parameter efficiency by limiting non-zero parameters, reducing computational and storage overhead, and accelerating inference. In knowledge swapping tasks, they enable the selective retention of crucial parameters for the current task while suppressing redundant ones, thereby preventing conflicts between new and existing knowledge. Additionally, parameter sparsification mitigates interference from irrelevant variables during learning and forgetting (similar idea in model pruning), allowing the model to focus on important information. Specifically, Lasso regularization penalizes the Frobenius norms of matrices, promoting group sparsification that selectively retains and forgets at the module level and maintains stability in low-level feature parameters when introducing new knowledge. Consequently, the model preserves new knowledge while maintaining the stability of existing representations. Overall, sparse constraints effectively manage parameters in knowledge swapping, enabling efficient adaptation to new tasks and maintaining previously acquired knowledge, thereby supporting scalable continual learning.

### 4.4. Training and Inference Protocol

In the **learning phase**, the objective is for the model to acquire new knowledge while retaining essential existing knowledge. Accordingly, the loss function for this phase is defined as follows.

$$\mathcal{L}_{\text{retain}} = \mathcal{L}(f(\boldsymbol{X}_r), \boldsymbol{Y}_r), \tag{5}$$

$$\mathcal{L}_{\text{learn}} = \mathcal{L}(f(\boldsymbol{X}_l), \boldsymbol{Y}_l), \tag{6}$$

$$\mathcal{L}_{\text{all}} = \mathcal{L}_{\text{retain}} + \beta\mathcal{L}_{\text{learn}} + \alpha\mathcal{L}_{\text{re}}, \tag{7}$$

where $(\boldsymbol{X}_r, \boldsymbol{Y}_r)$ denotes the data from the retention set, $(\boldsymbol{X}_l, \boldsymbol{Y}_l)$ denotes the data from the learning set, and $\alpha$ and $\beta$ are hyperparameters that balance the contributions of the different loss components.

In the **forgetting phase**, the goal is to eliminate specific knowledge while retaining both the original and newly acquired knowledge. This involves minimizing $\mathcal{L}(f(\boldsymbol{X}_r), \boldsymbol{Y}_r)$ and $\mathcal{L}(f(\boldsymbol{X}_l), \boldsymbol{Y}_l)$, while maximizing $\mathcal{L}(f(\boldsymbol{X}_f), \boldsymbol{Y}_f)$. However, directly maximizing the negative loss (i.e., minimizing $-\mathcal{L}(f(\boldsymbol{X}_f), \boldsymbol{Y}_f)$) can result in unbounded loss growth, leading to optimization instability. To address this issue, we introduce a boundary constraint (BND) to stabilize the loss. The final loss function for the forgetting phase is defined as:

$$\mathcal{L}_{\text{forget}} = \text{ReLU}(\text{BND} - \mathcal{L}(f(\boldsymbol{X}_f), \boldsymbol{Y}_f)), \tag{8}$$

$$\mathcal{L}_{\text{all}} = \mathcal{L}_{\text{retain}} + \mathcal{L}_{\text{learn}} + \beta\mathcal{L}_{\text{forget}} + \alpha\mathcal{L}_{\text{re}}, \tag{9}$$

where BND defines the forgetting boundary, and $(\boldsymbol{X}_f, \boldsymbol{Y}_f)$ represents the data from the forgetting set.

## 5. Experiments

### 5.1. Experimental Setup

All experiments are conducted on a hardware setup comprising 2×RTX 4090 GPUs, with the software environment configured as Python 3.12, PyTorch 2.5.1, and CUDA 12.4. The AdamW optimizer is employed for all training and forgetting phases.

For **image classification** tasks, the learning set includes: CUB-200-2011 (Wah et al., 2011), Oxford-IIIT Pet (Parkhi et al., 2012), RESISC45 (Cheng et al., 2017) and PlantVillage (Geetharamani & Pandian, 2019). Both the retention

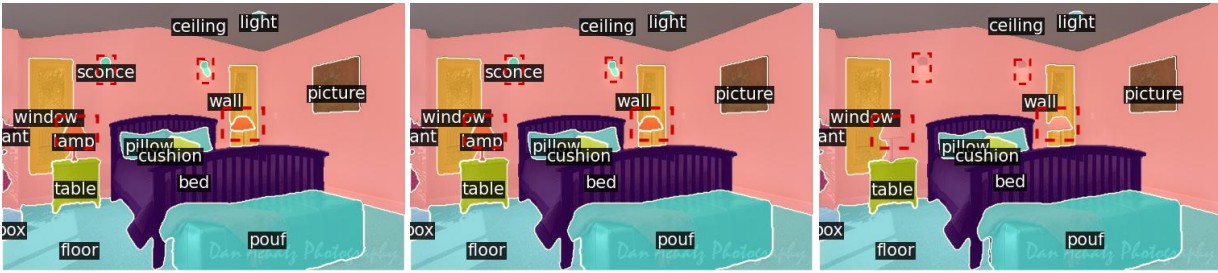

**The forgotten classes are *lamp* and *scope*. The learned class is *cow*. The retained classes are all remaining classes.**

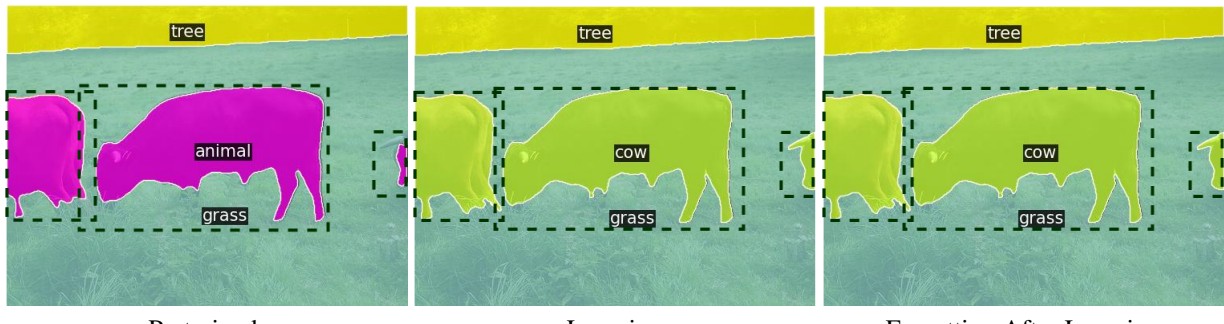

|  Pretrained  |  Learning  |  Forgetting After Learning  |

*Figure 5.* **Qualitative results on semantic segmentation.** The forgotten classes are marked with red dotted lines, and the learned class is marked with dark green dotted lines.

set and the forgetting set are selected from ImageNet-100. During the training phase, the hyperparameters are set to $\alpha = 0.05$ and $\beta = 0.2$, while BND $= 105$ in the forgetting phase. The classification performance is evaluated using accuracy as the primary metric.

For **object detection** tasks, the learning set consists of CUB-200-2011 and Stanford Dogs (Dataset, 2011). Both the retention set and the forgetting set are sourced from the COCO (Lin et al., 2014) dataset. The learning phase employs $\alpha = 0.01$ and $\beta = 0.9$, while the forgetting phase uses BND $= 15$, $\alpha = 0.01$, and $\beta = 0.2$. The detection capability is assessed using the mean Average Precision (mAP) metric.

For **semantic segmentation** tasks, the learning set includes: Pascal VOC (Hoiem et al., 2009), COCO, Oxford-IIIT Pet (Parkhi et al., 2012), and DeepGlobe Land (Demir et al., 2018). These datasets cover a diverse range of segmentation domains. The learning phase is configured with $\alpha = 0.01$ and $\beta = 0.9$ while the forgetting phase is set to BND $= 115$, $\alpha = 0.01$, and $\beta = 0.2$. The segmentation accuracy is evaluated using the mean Intersection over Union (mIoU) metric.

### 5.2. Classification Results

We use VIT-B16 model pretrained on ImageNet100 as the base model. As shown in Table 1, under the *Learning Before Forgetting* setting, the accuracy of the learning set consistently increases from approximately 0% to over 90%, demonstrating effective knowledge acquisition. Concur-

rently, the forgetting phase appears to be more straightforward, as the accuracy of the forgetting set decreases from approximately 80% to 0%, indicating successful forgetting. Besides, the retaining set can also be well preserved even though there exist limited negative impacts on the whole phase. We further evaluate the reverse order (*Learning After Forgetting*) in the same setting. We observe that although the accuracy of the forgetting set initially drops significantly during the forgetting phase, the subsequent learning phase induces substantial changes in the model's lower-level parameters. This renders the previously forgotten higher-level parameters ineffective, resulting in an increase in the accuracy of the forgetting set during the later stages. These findings confirm the validity of our hypothesis.

### 5.3. Semantic Segmentation Results

We select the pretrained Mask2Former(Cheng et al., 2021) model on ADE20K as the pretraining model. The results in Table 2 demonstrate that under the *Learning Before Forgetting* strategy, the mIoU of the retention set remains stable, ensuring effective memory retention. In contrast, the mIoU of the forgetting set decreases from 68.31% to nearly 0%, indicating complete forgetting. Concurrently, the learning set achieves a high mIoU, confirming successful knowledge acquisition. Figure 5 further illustrates this process, where the classes *lamp* and *sconce* are successfully forgotten and blended into the wall, while the learned class *cow* remains well retained even after the forgetting phase. In contrast, under the *Learning After Forgetting* setting (Table 2), al-

*Table 2.* **Semantic segmentation results on four datasets.** For each dataset, the learning set and forgetting set include randomly selected 5 classes. The retaining set includes all other classes from ADE20K.

| procedure | VOC | | | Oxford-pet | | | COCO | | | Deepglobe Land | | |
|---|---|---|---|---|---|---|---|---|---|---|---|---|
| | mIoU$_r$ ↑ | mIoU$_l$ ↑ | mIoU$_f$ ↓ | mIoU$_r$ ↑ | mIoU$_l$ ↑ | mIoU$_f$ ↓ | mIoU$_r$ ↑ | mIoU$_l$ ↑ | mIoU$_f$ ↓ | mIoU$_r$ ↑ | mIoU$_l$ ↑ | mIoU$_f$ ↓ |
| Start | 50.51 | 0 | 68.31 | 50.51 | 0 | 68.31 | 50.51 | 0 | 68.31 | 50.51 | 0 | 68.31 |
| $F$ | 50.36 | 0 | 2.26 | 50.61 | 0 | 3.48 | 50.24 | 0 | 2.41 | 50.66 | 0 | 3.65 |
| $F \rightarrow L$ | 50.7 | 85.45 | 49.42 | 50.28 | 59.45 | 53.67 | 51.03 | 90.87 | 54.47 | 50.38 | 54.33 | 60.25 |
| $F \rightarrow L \rightarrow F$ | 50.98 | 88.07 | 0.15 | 50.17 | 61.85 | 0.33 | 50.64 | 94.64 | 0.39 | 50.87 | 63.86 | 0.27 |
| $L$ | 50.2 | 84.97 | 60.67 | 48.92 | 62.21 | 65.5 | 50.54 | 93.46 | 61.85 | 50.96 | 59.8 | 64.61 |
| $L \rightarrow F$ | 50.57 | 85.43 | 0.12 | 49.87 | 69.55 | 0.08 | 50.35 | 93.36 | 0.71 | 49.54 | 58.34 | 0.4 |
| $L \rightarrow F \rightarrow L$ | 50.5 | 95.83 | 45.51 | 50.97 | 86.98 | 53.87 | 50.41 | 97.27 | 50.03 | 50.54 | 69.18 | 57.25 |

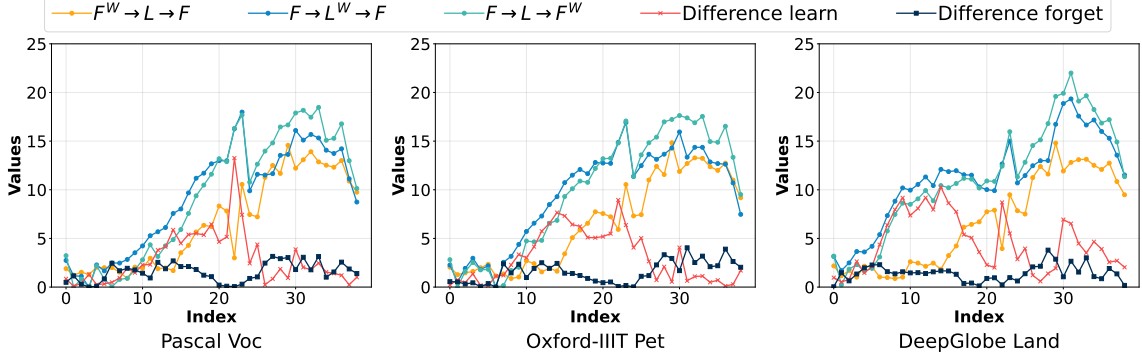

*Figure 6.* $\mathcal{L}_2$ norm of weights in $F \rightarrow L \rightarrow F$

though both the retention and learning sets perform well, the mIoU of the forgetting set increases after learning due to significant changes in low-level parameters, rendering previously tuned high-level forgetting parameters ineffective. Figure 7 highlights this issue, showing that *mountain*, which is initially erased and blended into *sand*, re-emerges after learning, demonstrating the instability of this approach.

### 5.4. Object Detection Results

For the object detection task, we use DINO (Zhang et al., 2022) pretrained on the COCO dataset as the base model. The forgetting set consists of five randomly selected classes: *person*, *teddy bear*, *toilet*, *bench*, and *bed*, while all remaining classes form the retention set. The learning sets consist of 5 classes: *Black-footed Albatross*, *Laysan Albatross*, *Sooty Albatross*, *Groove-billed Ani*, and *Brewer Blackbird* for CUB-200-2011 and *Chihuahua*, *Maltese Dog*, *Basset*, *American Staffordshire Terrier*, and *Norwich Terrier* for Stanford Dogs, respectively.

*Table 3.* Object Detection Results on COCO.

| procedure | Cub | | | Stanford Dogs | | |
|---|---|---|---|---|---|---|
| | mAP$_r$ ↑ | mAP$_l$ ↑ | mAP$_f$ ↓ | mAP$_r$ ↑ | mAP$_l$ ↑ | mAP$_f$ ↓ |
| Start | 55.4 | 0 | 44.5 | 55.4 | 0 | 44.5 |
| $F$ | 54.9 | 16.8 | 3.4 | 54.8 | 9.7 | 3.4 |
| $F \rightarrow L$ | 55 | 65.4 | 7.7 | 54.6 | 78.5 | 11.1 |
| $L$ | 55.3 | 64.3 | 37.6 | 54.5 | 67.5 | 37.5 |
| $L \rightarrow F$ | 55.5 | 62.2 | 0.5 | 55 | 80.1 | 0.6 |

The quantitative results in Table 3 demonstrate the effectiveness of our *Learning Before Forgetting* strategy, as the mAP of the retention set remains stable at around 55%, the for-

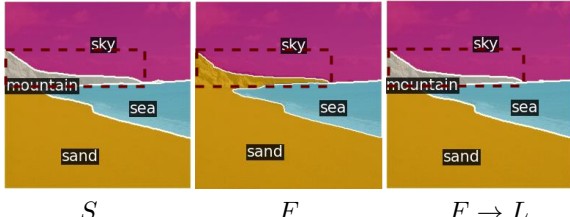

*Figure 7.* Semantic segmentation results for *Learning After Forgetting* on ADE20K.

getting set drops from 44.5% to below 1%, and the learning set improves from 0 to a satisfactory level, confirming the success of our approach. In contrast, it also shows that in the *Learning After Forgetting*($F \rightarrow L$) setting, the forgetting set retains a relatively high mAP, indicating ineffective forgetting. This supports our hypothesis that learning progresses from low-level to high-level features, while forgetting follows the opposite direction, from high-level to low-level features.

### 5.5. Insights and Discussion

To further validate the effectiveness of our strategy, we conduct an additional experiment involving a *Forget-Learn-Forget* sequence. As shown in Table 2, although the mIoU of the forgetting set remains high after the initial forgetting and subsequent learning phases, it is significantly reduced after the second forgetting phase. This result demonstrates the robustness of the *Learn Before Forget* strategy. The results $L \rightarrow F \rightarrow L$ further provide indirect evidence that

positioning the learning phase at the end influences the content that was previously forgotten. We analyze the $\mathcal{L}_2$ norm of model parameters at different stages, as illustrated in Figure 6. The red line represents parameter changes from $F$ to $F \rightarrow L$, while the black line indicates changes from $F \rightarrow L$ to $F \rightarrow L \rightarrow F$. The results show that learning-induced parameter changes are primarily concentrated in the early layers of the model, whereas forgetting-related changes are more prominent in the later layers. These observations align well with our previous findings.

## 6. Conclusion and Future Works

We proposed **Knowledge Swapping**, a novel task that enables selective regulation of model knowledge by achieving three goals: forgetting user-specified knowledge, retaining essential knowledge, and learning new knowledge. To accomplish this, we introduced a two-stage training strategy based on the *Learning Before Forgetting* principle, which decouples learning and forgetting to mitigate catastrophic forgetting effectively. We benchmark our *Learning Before Forgetting* with various experiments. However, our experiments also reveal that the difficulty of learning new knowledge for different categories and forgetting old knowledge varies. An interesting direction for future research is to explore and analyze the difficulty of forgetting specific knowledge and learning new knowledge of categories.

## Acknowledgements

This work has been supported by the New Cornerstone Science Foundation through the XPLORER PRIZE. This work was funded by the National Natural Science Foundation of China (No. 72188101 & No. 62472139 & No. 62302140 ) and the Fundamental Research Funds for the Central Universities (No. JZ2024HGTB0261 & No. JZ2024HGTA0178). The computation is completed on the HPC Platform of Hefei University of Technology.

## Impact Statement

This paper presents work whose goal is to advance the field of Machine Learning. There are many potential societal consequences of our work, none of which we feel must be specifically highlighted here.

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
