# OpenReview forum: "Knowledge Swapping via Learning and Unlearning"
_ICML.cc/2025/Conference — ICML 2025 poster_

### Official Review · Reviewer_ab73 · 2025-03-11

**Overall Recommendation:** 3

**Summary:**

1. This paper introduces Knowledge Swapping, a novel task designed to selectively regulate a pretrained model's knowledge by enabling the forgetting of user-specified information while retaining essential knowledge and acquiring new knowledge simultaneously.

2. The authors propose a two-stage training strategy based on the "Learning Before Forgetting" principle, which decouples learning and forgetting processes to effectively mitigate catastrophic forgetting and achieve better performance in knowledge regulation.

3. Comprehensive experiments across various tasks including image classification, object detection, and semantic segmentation demonstrate the effectiveness of the proposed approach, showing significant improvements in retaining essential knowledge, forgetting specified content, and learning new information compared to alternative approaches.

**Claims And Evidence:**

1. Introduction of Knowledge Swapping as a novel task: This claim is well-supported. The authors clearly define the task, distinguish it from related approaches like continual learning and machine unlearning, and provide a mathematical formulation of the objectives. The comparison to existing tasks (Figure 1) further strengthens this claim.

2. Discovery of the directional contrast between learning and forgetting: This claim is supported by experimental evidence. The authors conduct experiments analyzing parameter changes during learning and forgetting phases across different layers of neural networks. Figures 2 and 4 show that learning affects later layers (high-level features) while forgetting affects earlier layers (low-level features).

3. Effectiveness of the "Learning Before Forgetting" strategy: This claim is well-supported. The authors present experimental results across multiple tasks (image classification, object detection, semantic segmentation) showing that the Learning Before Forgetting approach consistently achieves better performance in terms of retention, forgetting, and learning objectives. Tables 1-3 demonstrate superior results for their proposed method compared to the reverse approach.

4. Benchmark framework using LoRA with group sparse regularization: This claim is supported by technical details and experimental results. The framework is shown to work as intended, enabling efficient and effective knowledge regulation while maintaining parameter efficiency.

5. Variation in difficulty of learning and forgetting across categories: This claim is somewhat speculative. While it may be a reasonable hypothesis based on their work, the authors do not provide specific quantitative analysis or dedicated experiments to support this claim. It is mentioned more as a future research direction rather than a firmly established finding.

**Essential References Not Discussed:**

NO

**Experimental Designs Or Analyses:**

1. The authors evaluate their method across three different computer vision tasks: image classification, object detection, and semantic segmentation. This demonstrates the generalizability of their approach across different types of vision problems.

2. The authors specifically compare their proposed "Learning Before Forgetting" strategy against the reverse approach ("Forgetting Before Learning") across all tasks. This direct comparison effectively demonstrates the superiority of their proposed method.

3. They provide both quantitative results (tables) and qualitative results (figures) to comprehensively evaluate their method's effectiveness.
The qualitative results help visualize the practical implications of their findings.

**Methods And Evaluation Criteria:**

1. Learning Before Forgetting Strategy:
1.1 This two-stage approach is well-motivated by the authors' analysis of how learning and forgetting affect different layers of neural networks. Their experiments showing that learning affects higher-level semantic features while forgetting impacts lower-level features provide a logical basis for sequencing these processes.
1.2 The strategy effectively decouples the learning and forgetting processes, which helps mitigate catastrophic forgetting and allows for more controlled knowledge regulation.

2. LoRA with Group Sparse Regularization:
2.1 Using Low-Rank Adaptation (LoRA) for fine-tuning is appropriate as it allows efficient parameter updates while preserving pretrained knowledge. This choice makes sense for vision tasks where Transformers have become standard architectures.
2.2 The group sparse regularization approach is suitable for selectively retaining and forgetting knowledge at the module level within the Feed-Forward Network (FFN) modules. This method enables targeted parameter control without excessive computational overhead.

3. Boundary Constraint for Forgetting:
3.1 The introduction of a boundary constraint (BND) in the forgetting phase addresses potential optimization instability issues that could arise from directly maximizing the negative loss. This technical refinement demonstrates thoughtful consideration of implementation details.

**Other Comments Or Suggestions:**

No

**Other Strengths And Weaknesses:**

1. While the paper demonstrates the effectiveness of their overall approach, there could be more extensive ablation studies to understand the contribution of individual components (e.g., the impact of different regularization strengths, the importance of sparse constraints).

2. The hyperparameters (α, β, BND) are set based on what worked well in their experiments, but there's limited analysis of how sensitive the method is to these choices. A more thorough exploration of the hyperparameter space could strengthen the robustness of their conclusions.

3. The experiments primarily focus on Transformer-based models (VIT-B16, Mask2Former). While this is appropriate given the current trends in vision research, it would be valuable to see how the method performs on other architectures like CNNs.

4. While the method is shown to be effective, there's limited discussion of computational efficiency compared to alternative approaches.
Information about training time, memory usage, or parameter efficiency could provide additional insights into the practicality of their method.

**Questions For Authors:**

1. While the paper demonstrates the effectiveness of their overall approach, there could be more extensive ablation studies to understand the contribution of individual components (e.g., the impact of different regularization strengths, the importance of sparse constraints).

2. The hyperparameters (α, β, BND) are set based on what worked well in their experiments, but there's limited analysis of how sensitive the method is to these choices. A more thorough exploration of the hyperparameter space could strengthen the robustness of their conclusions.

3. The experiments primarily focus on Transformer-based models (VIT-B16, Mask2Former). While this is appropriate given the current trends in vision research, it would be valuable to see how the method performs on other architectures like CNNs.

4. While the method is shown to be effective, there's limited discussion of computational efficiency compared to alternative approaches.
Information about training time, memory usage, or parameter efficiency could provide additional insights into the practicality of their method.

**Relation To Broader Scientific Literature:**

1. Curriculum Learning Parallels: The proposed strategy shares conceptual similarities with curriculum learning (Bengio et al., 2009), where the order of learning tasks can significantly impact model performance. Just as curriculum learning orders tasks from simple to complex, the "Learning Before Forgetting" approach sequences knowledge regulation in a way that leverages the natural progression of feature learning.

2. Mitigating Catastrophic Forgetting: This approach addresses catastrophic forgetting more effectively than traditional methods by decoupling learning and forgetting. Unlike regularization-based methods that attempt to balance retention and new learning simultaneously (Li & Hoiem, 2017), the sequential approach allows for more controlled knowledge regulation.

**Theoretical Claims:**

In this paper, the authors don't present formal mathematical proofs for their claims. Instead, they rely on empirical evidence from experiments to support their theoretical insights.

---

> ### Author Rebuttal · Authors · 2025-04-01
>
> We sincerely thank Reviewer ab73 for the valuable comments. Reviewer ab73 notes that our **claims are well-supported by experimental evidence and technical details**, acknowledges that **Tables 1–3 demonstrate superior results** for our proposed method compared to the reverse approach, commends our two-stage approach as **well-motivated by our analysis** of how learning and forgetting affect different layers of neural networks, and highlights the **generalizability of our method across various vision problems**. We address the main concerns as follows:
>
> >**Additional results using other architectures, such as CNNs.**
>
> Good point.  We conduct additional experiments using the ResNet-18 architecture to learn five new classes and forget five original classes. These additional results consistently support the key insight of "Learning before Forgetting.”
> |Procedure||Cub|||Oxford-pet||
> |---------|---|---|---|----------|---|---|
> |         |Acc$_r$↑|Acc$_l$↑|Acc$_f$↓|Acc$_r$↑|Acc$_l$↑|Acc$_f$↓|
> |Start|77.32|0|68.00|77.32|0|68.00|
> |F|77.874|0|3.60|78.52|0|4.0|
> |F→L|75.13|50|16.40|76.12|54.80|11.60|
> |L|76.48|51.19|58.80|78.16|58.4|59.6|
> |L→F|76.88|73.81|0.0|76.21|81.60|0.4|
>
>
> |Procedure||Resisc45|||Plantvillage||
> |---------|:-------:|:-------:|:-------:|:-------:|:-------:|:-------:|
> |         |Acc$_r$↑|Acc$_l$↑|Acc$_f$↓|Acc$_r$↑|Acc$_l$↑|Acc$_f$↓|
> |Start|77.32|0|68.00|77.32|0|68.00|
> |F|79.05|0|3.2|78.04|0|2.4|
> |F→L|76.08|77.60|6.0|73.43|70.53|6.80|
> |L|75.95|72.2|54.0|71.83|87.95|51.20|
> |L→F|77.03|95.0|0.4|77.20|97.20|1.60|
>
>
> >**More ablation studies about individual components, such as the impact of different regularization strengths and the importance of sparse constraints.**
>
> Thank you for this suggestion. The table below summarizes the performance metrics on the CUB dataset when excluding sparse constraints. Metrics are provided under various conditions (default settings: $\alpha_l = \alpha_f = 0.01$):
>
> |Condition|CUB mAP$_r$↑|CUB mAP$_l$↑|CUB mAP$_f$↓|
> |---------|------------|------------|------------|
> |$\alpha_l$=0, $\alpha_f$=0|55.4|60.3|0.4|
> |$\alpha_l$=0, $\alpha_f$=0.01|55.5|61.2|0.3|
> |$\alpha_l$=0.01, $\alpha_f$=0|55.3|61.3|0.5|
> |$\alpha_l$=0.01, $\alpha_f$=0.01|55.5|62.2|0.5|
>
> >**More ablation studies about  hyperparameters ($\alpha$, $\beta$, BND).**
>
> Below we provide detailed ablation results illustrating the impact of various hyperparameters on detection performance using the CUB dataset as the learning set.
>
> **Effect of BND (default: BND=15)**
>
> |BND|CUB mAP$_r$↑|CUB mAP$_l$↑|CUB mAP$_f$↓|
> |-|-|-|-|
> |5|55.7|64.5|30.5|
> |15|55.5|62.2|0.5|
> |25|55.6|48.4|5.4|
> |50|55.2|44.3|8.1|
> **Effect of $\beta_{learn}$ (default: $\beta_{learn}=0.9 $)**
>
> |$\beta_{learn}$|CUB mAP$_r$↑|CUB mAP$_l$↑|CUB mAP$_f$↓|
> |-|-|-|-|
> |50|55.4|55.9|1.0|
> |10|55.3|61.4|0.7|
> |2|55.4|61.8|0.6|
> |0.9|55.5|62.2|0.5|
> |0.5|55.5|60.5|0.3|
> |0.1|55.7|48.9|0.2|
> |0|55.5|40.7|1.8|
>
> **Effect of $\beta_{forget}$ (default: $\beta_{forget}=0.2$)**
> |$\beta_{forget}$|CUB mAP$_r$↑|CUB mAP$_l$↑|CUB mAP$_f$↓|
> |-|-|-|-|
> |1|55.4|57.7|0.2|
> |0.5|55.4|60.2|0.6|
> |0.2|55.5|62.2|0.5|
> |0.1|55.7|63.8|0.5|
> |0|55.6|64|35.7|
>
> **Effect of $\alpha$ (default: $\alpha_l = \alpha_f = 0.01$)**
> |$\alpha$|CUB mAP$_r$↑|CUB mAP$_l$↑|CUB mAP$_f$↓|
> |-|-|-|-|
> |0.001|55.4|61.7|0.4|
> |0.01|55.5|62.2|0.5|
> |0.1|55.4|49.4|0.4|
> |1|55.2|46.1|0.2|
>
> >**Computational efficiency discussion.**
>
> Thank you for highlighting this aspect. The following table provides a comparative analysis of training time, inference speed per image, and the number of trainable parameters for various model architectures. It demonstrates the trade-offs among training duration, inference efficiency, and parameter efficiency:
> |Model|Training Time|Inference Time (per image)|Trainable Parameters|
> |-----|-------------|--------------------------|--------------------|
> |ResNet-18 (all)|0.3 h|0.0007 s|11.2 M|
> |Vit16_B (LoRA)|0.5 h|0.0035 s|0.74 M|
> |Dino (LoRA)|4 h|0.084 s|1.8 M|
> |Mask2former (LoRA)|4 h|0.0037 s|1.4 M|
>
> **Should further clarification or additional details be necessary, we welcome further discussion during the rebuttal period.**

---

### Official Review · Reviewer_tJYm · 2025-03-11

**Overall Recommendation:** 3

**Summary:**

This paper introduces a new task called Knowledge Swapping, which aims to regulate the knowledge of a pretrained model by optimizing three objectives: forgetting user-specified knowledge, retaining core pretrained knowledge, and simultaneously learning new knowledge. The authors empirically demonstrate that learning new knowledge before forgetting specified knowledge leads to better results than the reverse order.

**Claims And Evidence:**

The manuscript is based on the claim that in the Learning then Forgetting sequence, most parameter updates occur in the latter layers of the neural network, while in the Forgetting then Learning sequence, changes are concentrated in the earlier layers.
However, in my opinion the empirical study presented in Figure 2 does not clearly validate such a claim. Specifically, the observed parameter norms appear similar regardless of whether the Learning or Forgetting phase comes first. Moreover, the value of weight norms alone may not be a suitable metric to evaluate the extent to which different layers are affected, as it does not directly capture changes in feature representations or their semantic hierarchy. I suggest using more established measures of change across layers, such as CKA as in [1,2], to make a stronger argument.

[1]:  Boschini, Matteo, et al. Transfer without forgetting. In ECCV 2022.
[2]: Ramasesh, V. V., et al. Anatomy of catastrophic forgetting: Hidden representations and task semantics. In ICLR 2020.

**Essential References Not Discussed:**

N/A

**Experimental Designs Or Analyses:**

The experimental design is overall valid but lacks an evaluation of more than three sequential learning and forgetting phases. Since the main experiments only considers a single learning and forgetting cycle, it remains unclear whether the observed forgetting effects stem from the order of these phases, as claimed by the authors, or from the model’s ability to set to 0 the LoRA parameters, thus returning to the pre-training configuration.

**Methods And Evaluation Criteria:**

Since the manuscript is positioned within a continual learning context, I believe it would be valuable to include experiments involving more than three sequential learning and forgetting phase (e.g., alternating learn→forget→learn→forget->learn).

Furthermore, the forgetting sets used in the experiments are predefined subsets of the pretraining data. It would be interesting to explore the forgetting of emergent knowledge not included in the pretraining data, as such knowledge cannot be forgotten by zeroing out the learned LoRA matrices.

**Other Comments Or Suggestions:**

N/A

**Other Strengths And Weaknesses:**

Strengths:
1) The manuscript introduces a practical and novel task (Knowledge Swapping) with real-world applications (e.g., privacy compliance, model adaptation).
2) The proposed approach is based on the insight that incremental learning progresses from low-level to higher-level semantic features, which offers an actionable strategy for dynamic model adaptation.

Weaknesses:
1) The claim that incremental learning follows a progression from low-level to high-level features is not convincingly demonstrated, as the experiments do not provide clear empirical evidence to support it.
2)  Experiments do not reflect continual learning’s iterative nature or emergent forgetting scenarios. While they show the effectiveness of the proposed strategy, they do not allow for direct comparison with existing continual learning approaches. In particular, I believe the results observed during the forgetting phase may be attributed to the model's ability to effectively nullify the contributions of LoRA weights, rather than reflecting the model's inherent capacity to forget a given task. This I believe is also supported by Tab. 2, where it seems that the only important thing in the evaluated scenario is for the forgetting phase to happen last in the sequence.

**Questions For Authors:**

See section on "Claims And Evidence" and "Methods And Evaluation Criteria"

**Relation To Broader Scientific Literature:**

The original claim regarding the order of learning and forgetting would be a substantial contribution.

**Theoretical Claims:**

N/A

---

> ### Author Rebuttal · Authors · 2025-04-01
>
> We sincerely thank Reviewer tJYm  for the insightful and constructive comments. Reviewer tJYm finds that "Knowledge Swapping is a **practical and novel task**", “the experimental design is overall **valid**”, and "learning before forgetting would be a **substantial contribution**". We address the main concerns below.
> >**Reviewer tJYm  think (1) weight norms in Fig. 2 do not convincingly validate the key claim; (2) more metrics like CKA are suggested to make a stronger argument.**
>
> Thanks. (1) We would like to clarify potential confusion regarding Fig. 2. Specifically, in the $L^W→F$ (Fig.2(a)), the shallow layer weight norms clearly increase significantly during the learning phase compared to those in the $F^W→L $ (Fig.2(b)). Given that we initialize the weights uniformly using Kaiming initialization, resulting in an average initial norm of approximately 1.12, the observed increase in shallow-layer norms in Fig. 2(a) implies substantial updates occurring at shallow layers during learning (thereby subsequently impacting the middle-to-deep layers).  Conversely, the elevated norms observed in Fig. 2(b), when the forgetting phase precedes learning, occur primarily in middle-to-deep layers. **This difference indicates that when learning precedes forgetting, modifications mainly affect higher-level semantic information, whereas in the forgetting-first scenario, substantial updates are more prominent in shallow layers.** We hope this clarification adequately addresses the reviewer's concern.
>
> (2)  Good suggestion!  We have conducted additional validation experiments using the CKA metric as recommended. The detailed results   are available (Fig. S1 and Fig. S2) at the anonymous link (https://anonymous.4open.science/r/rebuttal-C764/figures.pdf) and further corroborate our original findings.
>
> **Should additional clarification be required, we would gladly engage in further discussion during the rebuttal period.**
>
> >**Reviewer tJYm  suggests more sequential $L$ and $F$ phase.**
>
> Good point. We have explored sequential phases of learning and forgetting as reported in Tab.2 (F→L→F and L→F→L). We have now extended these experiments by adding further cycles, as shown in the updated results below:
>
> |||VOC|||Oxford-pet||
> |-|-|-|-|-|-|-|
> ||mIoU$_r$↑|mIoU$_l$↑|mIoU$_f$↓|mIoU$_r$↑|mIoU$_l$↑|mIoU$_f$↓|
> |Start|50.51|0|68.31|50.51|0|68.31|
> |F|50.36|0|2.26|50.61|0|3.48|
> |F→L|50.70|85.45|49.42|50.28|59.45|53.67|
> |F→L→F|50.98|88.07|0.15|50.17|61.85|0.33|
> |F→L→F→L|51.28|96.21|40.95|50.84|88.86|45.94|
> |F→L→F→L→F|50.43|94.60|1.90|50.54|88.49|0.25|
> |L|50.20|84.97|60.67|48.92|62.21|65.50|
> |L→F|50.57|85.43|0.12|49.87|69.55|0.08|
> |L→F→L|50.50|95.83|45.51|50.97|86.98|53.87|
> |L→F→L→F|50.43|93.19|1.09|50.20|88.38|1.06|
> |L→F→L→F→L|50.51|97.50|33.28|51.03|91.12|43.35|
>
> The expanded experimental results continue to support our claim: concluding with  F→L makes forgetting significantly more challenging,  whereas ending with L→F effectively alleviate this. Additionally, multiple cycles appear to contribute to incremental performance improvements.
> >**Reviewer tJYm  questions whether the observed forgetting results reflect genuine forgetting or merely the model's ability to nullify LoRA weights.**
>
> Thanks. We respectfully provide an alternative interpretation based on the following considerations:
> (1) Our "knowledge swapping" task explicitly requires the model to forget knowledge originally learned during pretraining. Merely nullifying LoRA parameters (setting them to 0) would return the model to its pretraining state without achieving actual forgetting of previously learned knowledge.
>
> (2) Additionally, we conduct further experiments using CNN-based full tuning (ResNet-18). We set the model to learn 5 new classes and forget 5 original classes. These results still demonstrate the validity of our key insight. (Please see the first response for R#ab73).
>
> **Should further clarification be necessary, we would welcome additional discussion during the rebuttal period.**
> >**Reviewer tJYm  suggests to explore more results  about the forgetting of emergent knowledge not included in the pretraining data.**
>
> Thanks. We additionally conduct segmentation experiments on the COCO dataset, where "l5f1" denotes learning 5 new classes and forgetting 1 new class (with other settings similar), none of which are included in the pretraining. Interestingly, the accuracy for the forgotten class dropped to 0. This indicates that forgetting emergent knowledge (previously unknown during pretraining) can be readily achieved by nullifying specific LoRA parameters. However, our primary scenario involves forgetting previous knowledge (distinct from "emergent knowledge" as defined in Sec. 3.1). For detailed results and discussions, please refer to the previously response.
> ||mIOU$_r$↑| mIOU$_l$↑|mIOU$_f$↓|
> |---------|--------------|--------------|--------------|
> |l5f1|50.36|93.78|0|
> |l4f2|50.68|96.78|0|
> |l3f3|49.98|98.03|0|
> |l2f4|50.87|98.88|0|
> |l1f5|50.17|98.05|0|

---

### Official Review · Reviewer_3jJx · 2025-03-13

**Overall Recommendation:** 4

**Summary:**

This paper proposed Knowledge Swapping, a novel task designed to regulate knowledge of a pretrained model selectively. Meanwhile, this paper uncovers that incremental learning progresses from low-level to higher-level semantic features, whereas targeted forgetting begins at high-level semantics and works downward. Therefore, the paper achieves knowledge swapping by the sequential learning-then-forgetting principle. Comprehensive experiments on various tasks like image classification, object detection, and semantic segmentation validate the effectiveness of the proposed strategy.

**Claims And Evidence:**

The claims of this paper are clear and supported by convincing evidence. The claims are summarized as:
1)	Knowledge Swapping is an interesting and novel task.
2)	The incremental learning progresses from low-level to higher-level semantic features, whereas targeted forgetting begins at high-level semantics and works downward. This is the motivation for how to design effective knowledge-swapping procedures.
3)	Comprehensive experiments on various tasks like image classification, object detection, and semantic segmentation validate the effectiveness of the proposed strategy.

**Essential References Not Discussed:**

None.

**Experimental Designs Or Analyses:**

Yes, the theoretical claims in the paper are valid. The experimental designs and analyses have been carefully checked, covering three tasks: image classification, object detection, and semantic segmentation. The results demonstrate the effectiveness of the proposed method, and the analysis is reasonable, further supporting the correctness of the claims.

**Methods And Evaluation Criteria:**

The proposed method appears to be reasonable based on the following key aspects.
(1). Empirical Justification of Learning and Forgetting Strategies (Section 3.2).
-The paper systematically investigates the impact of learning-before-forgetting and forgetting-before-learning strategies by analyzing parameter changes across multiple image segmentation tasks.
-By comparing these strategies under controlled settings, the method provides empirical evidence supporting the claim.
(2). Logical and Coherent Model Design (Section 4).
-The model's formulation aligns well with the problem setting, ensuring that each component has a justified role in improving performance.
-The theoretical reasoning provided in this section lays a solid foundation for the model’s expected behavior.
(3). Generalization and Robustness Considerations.
-The proposed method is tested on multiple datasets and task, and the findings are consistent, it implies that the approach is not overly specialized for a single case.
-The method’s adaptability to different settings (e.g., segmentation, classification) further validates its broader applicability.
(4). Minimal Unjustified Assumptions
-The paper does not seem to rely on overly strong assumptions that could limit its real-world applicability.

**Other Comments Or Suggestions:**

Suggestion: Add more examples in Figures 5 and 7 to make the experiments more comprehensive and the paper more convincing.

**Other Strengths And Weaknesses:**

(1). The writing of this draft is clear and content of each section is well-structured.
(2). Knowledge Swapping is a newly defined task that introduces a novel and intriguing concept. The proposal of this task expands the scope of deep learning and holds significant potential for advancing the industry.
(3). The experimental design is robust, incorporating various tests and analyses that provide compelling evidence to support the conclusions.

**Questions For Authors:**

As the authors mentioned in the limitations, the difficulty of learning and forgetting different types of knowledge varies. Could you discuss this further and suggest potential directions for future research?

**Relation To Broader Scientific Literature:**

(1). Knowledge Swapping is a good task. The concept of Knowledge Swapping introduces a novel approach to balancing learning new tasks while selectively forgetting less important or sensitive prior knowledge. The idea of Knowledge Swapping differs from existing continual learning and machine unlearning.
(2). The discovery of “learning-before-forgetting” is interesting and novel. This further provides a guidance for related researches.
(3). I think this work could be applied to various existing large-model-based works. For example, Privacy-Preserving AI and Federated Learning, AI Model Auditing and Compliance, etc.

**Theoretical Claims:**

Yes, Both claims seem reasonable given the paper’s empirical results and theoretical grounding.

---

> ### Author Rebuttal · Authors · 2025-04-01
>
> We thank Reviewer **3jJx** for the valuable comments. Reviewer 3jJx appreciates that "**Knowledge Swapping is a good task**," "is **interesting and novel**," "**well-structured**," represents a "**novel and intriguing concept**," and is "**robust**." Below, we provide detailed responses addressing the remaining concerns.
> >**Reviewer 3jJx asks for more results in Fig.5 & Fig.7.**
>
> Thank you for this suggestion. We have included additional related results (**Fig. S3 and Fig. S4**) in the supplementary material accessible via the anonymous link (https://anonymous.4open.science/r/rebuttal-C764/figures.pdf). These results will also be included explicitly in our final manuscript.
>
> >**Reviewer 3jJx suggests providing further discussion about the limitations of the current approach, specifically regarding the varying difficulty in learning and forgetting different types of knowledge, and outlining potential future directions.**
>
> We greatly appreciate Reviewer 3jJx’s insightful recommendation to elaborate on limitations. Our experiments have demonstrated that the difficulty in acquiring new knowledge and the ease of forgetting existing knowledge indeed vary significantly across different categories. Investigating these variations presents a meaningful future research direction. Specifically, exploring and characterizing the complexity associated with different knowledge categories can reveal critical insights. **One promising approach is to incorporate uncertainty-based assessment methods, as discussed by **[R1]**, to better evaluate model confidence and quantify these complexities**. Employing uncertainty estimation can further elucidate the underlying mechanisms influencing learning and forgetting within our proposed framework. Ultimately, this line of inquiry may foster the development of more robust, targeted, and efficient strategies in future research.
>
> **References:**
>
> [R1] Gawlikowski, Jakob, et al. "A survey of uncertainty in deep neural networks." *Artificial Intelligence Review*, 2023.

---

### Decision · Program_Chairs · 2025-05-01

**Decision:**

Accept (poster)

**Comment:**

This paper introduces Knowledge Swapping as a task, motivates learning then forgetting through looking at high and low-level features and how they change, and provides experiments across a variety of tasks.

Reviewers mostly agree that this is a good paper: the task of Knowledge Swapping is interesting, novel and well-motivated; learning then forgetting is well-motivated; experiments are performed across many different tasks and sufficiently.

The authors added experiments in the rebuttal about increased number of tasks (like in a continual learning setting), ablations on hyperparameters, and more. I encourage the authors to add these to the main paper (or appendices), including the ablation results.